# Controlling for Structural Changes in the Workforce Influenced Occupational Class Differences in Disability Retirement Trends

**DOI:** 10.3390/ijerph16091523

**Published:** 2019-04-30

**Authors:** Svetlana Solovieva, Taina Leinonen, Kirsti Husgafvel-Pursiainen, Antti Kauhanen, Pekka Vanhala, Rita Asplund, Eira Viikari-Juntura

**Affiliations:** 1Finnish Institute of Occupational Health, 00032 Helsinki, Finland; taina.leinonen@ttl.fi (T.L.); kirsti.husgafvel-pursiainen@ttl.fi (K.H.-P.); eira.viikari-juntura@ttl.fi (E.V.-J.); 2Research Institute of the Finnish Economy, 00100 Helsinki, Finland; antti.kauhanen@etla.fi (A.K.); pekka.vanhala@etla.fi (P.V.); rita.asplund@etla.fi (R.A.)

**Keywords:** mental disorders, musculoskeletal diseases, pension, propensity score matching, register study, work disability

## Abstract

We explored occupational class differences in disability retirement trends accounting for structural changes in the workforce induced by the recent economic crisis and the following economic stagnation. Using nationwide register data on the general Finnish population aged 30–59 years, we examined trends in disability retirement due to all causes, musculoskeletal diseases, and mental disorders in 2007, 2010, and 2013. Applying propensity score (PS) matching to control for bias induced by structural changes in the workforce over time, we obtained 885,807 matched triplets. In the original study population, all-cause and cause-specific disability retirement declined between 2007 and 2013 for most occupational classes. In the matched study population, the disability retirement among skilled and unskilled manual workers sharply increased in 2010 and then declined in 2013. PS matching considerably attenuated the decline in disability retirement, particularly between the years 2007 and 2010. In general, the differences in disability retirement between both skilled and unskilled manual workers and upper-level non-manual employees widened during the period of economic stagnation. In occupational epidemiology, structural changes in the workforce should be accounted for when analysing trends in ill-health. Controlling for these changes revealed widening occupational class differences in disability retirement during the period of economic stagnation.

## 1. Introduction

The working-age population is globally projected to shrink by 9% over the next 50 years [1]. Despite overall improvements in population health, the employment rate is relatively low, being on average around 70% in European countries [2]. Disability retirement is the most common pathway to prematurely terminate working life and results in the largest share of economic cost to society [3,4]. 

Challenges related to early exit from employment become more pronounced during an economic crisis. The recent global economic downturn led to severe changes in the labour market, characterized by increases in unemployment, underemployment and job insecurity, a reduction of certain types of jobs, and a sharp decline in the employment rate [5]. The population groups that were particularly affected by the crisis include both young and old people, less educated and low-skilled workers, as well as people with health problems. The crisis also accelerated ongoing changes in demographics and in the composition of the workforce. Technological progress and globalization have decreased the share of middle-paying occupations [6], and economic downturns have shrunk the share of low-wage workers [7] as well as low-paid jobs [8]. 

A recent systematic review found the most consistent evidence for the effect of the economic crisis on mental health deterioration and an increased risk of suicide [9]. However, the crisis did not reverse the trend of decreasing overall mortality. In the beginning of the crisis, a declining trend of unemployment was matched in many OECD (The Organisation for Economic Co-operation and Development) countries by an increasing trend in disability, suggesting that unemployed people were increasingly seen as incapable of working [10]. Furthermore, the health effects of the crisis are likely to be heterogeneous across age, gender, and type of morbidity.

Previous research showed that disability retirement is unequally distributed across population groups with major differences by age, gender, geographical region, and socioeconomic position [11,12,13,14,15,16,17,18,19]. Studies on occupation-based socioeconomic differences in disability retirement have typically used three categories: upper-level non-manual employees, lower-level non-manual employees, and manual workers. Understanding the variation in disability retirement across more specific occupational classes could reveal possibilities for more targeted prevention of exit from paid employment. However, such information is lacking. Furthermore, the impact of the economic crisis on occupation-specific trends in disability retirement has not been explored.

A complex interplay of within- and between-occupation compositional changes may lead to changes in the association of occupation with disability retirement. Recent findings from Finland report a downward trend in disability retirement between 2000 and 2009 in each socioeconomic class [19]. The validity of crude trend estimates is, however, sensitive to the comparability of the underlying population in different time periods, and may be weakened as a result of changes in the composition of the population and a reduction of the workforce (non-random attrition). The compositional changes in the workforce affect those with health problems in particular and increase the probability of ill-health based selection out of employment. An increase in such non-random attrition may over time ultimately affect the overall incidence of disability retirement and exaggerate the declines in the trend. To reduce or eliminate non-random attrition bias in trend analyses, propensity score (PS) matching [20] can be used to estimate different overall effects in differently defined populations [21]. However, this approach has so far not been used in trend analysis. 

Finland has one of the world’s most comprehensive social security systems with fairly generous pensions as well as sickness and unemployment benefits. Since 2004, several reforms have been introduced, aiming to reduce the system’s financial load by enhancing work participation via prevention of permanent full disability retirement. These include, among others, renovation of the pension accrual to better reflect the earnings of the entire work career, easier access to vocational rehabilitation, as well as the introduction of a partial sickness benefit and the ensuing widening of the criteria for its use. 

Finland was heavily hit by the economic crisis, particularly in 2009, when the gross domestic product (GDP) growth turned negative. After weak GDP growth in 2011, the growth turned again on a downward path, which lasted until late 2015 [22]. During the peak of the economic recession in 2009, we observed a temporary decrease in full sickness absence due to musculoskeletal diseases, particularly among manual workers in the manufacturing sector, that is, in a segment of wage earners that are known to have been hit hard by the recession [23]. 

Utilizing nationwide administrative registers, we explored occupation-specific disability retirement trends, taking into account structural changes in the workforce induced by the recent economic crisis and the following economic stagnation. For this, we compared the years 2007 and 2013 with 2010, applying PS matching. Furthermore, we investigated changes over time in occupational class differences in disability retirement. 

## 2. Material and Methods

### 2.1. Study Population

The study base consisted of three longitudinal cohorts, each including a 70% nationally representative random sample of the working-age general population living in Finland on the last day of the year 2004 (first cohort, *N* = 2,550,446), 2007 (second cohort, *N* = 2,582,752), and 2010 (third cohort, *N* = 2,617,963). The share of 30–59-year-old persons in the three cohorts was 60.2% (*N* = 1,536,101), 58.2% (*N* = 1,502,239), and 57.0% (*N* = 1,491,676), respectively. The data included information on earnings-related pensions from the Finnish Centre for Pensions (FCP), national pensions obtained from the Social Insurance Institution of Finland (SII), and sociodemographic factors obtained from the Finnish Longitudinal Employer‒Employee Data (FLEED) of Statistics Finland.

Register information was searched for the first cohort between 1 January 2005 and 31 December 2007 (observational year 2007); for the second cohort between 1 January 2008 and 31 December 2010 (observational year 2010); and for the third cohort between 1 January 2011 and 31 December 2013 (observational year 2013). Persons aged 30–59 years and who were employed according to their main economic activity at baseline (i.e., on the last day of the year preceding each observational year) were eligible for the study. We excluded persons with missing occupational title at baseline (<1%). 

Our final study population consisted of 1,157,130, 1,119,929, and 1,114,789 individuals for the observational years 2007, 2010, and 2013, respectively. 

### 2.2. Occupational Class

Information on status in employment (employee or self-employed) and occupation held at baseline was obtained from the FLEED. The occupations were classified using the Classification of Occupations 2001 by Statistics Finland, which is based on the International Standard Classification of Occupations (ISCO-88). Six occupational classes were formed: (1) upper-level non-manual employees, (2) lower-level non-manual employees, (3) skilled manual workers, (4) unskilled manual workers, (5) self-employed farmers and agricultural workers, and (6) other self-employed.

### 2.3. Disability Retirement

All Finnish residents with a chronic illness, disability, or injury that have been verified by a physician with a medical certificate and evaluated as causing considerable and long-lasting (about one year) decreased work ability are entitled to a disability pension. Before that, sickness allowance is usually paid for a maximum of 300 working days [24]. 

The FCP and SII registers provide information on all disability retirement events with their primary and secondary diagnoses as classified according to the International Statistical Classification of Diseases and Related Health Problems, Tenth Revision (ICD-10, Finnish version of ICD-classification 1996). The outcome used in this study was the annual cumulative incidence of full-time (temporary or permanent) disability retirement. In addition to all causes, the two largest main diagnostic groups, including diseases of the musculoskeletal system and connective tissue (M00–M99) as well as mental and behavioural disorders (F00–F99), were examined separately. Among the remaining diagnostic groups, diseases of the circulatory system (I00–I99), neoplasms (C00–C99), injuries (S00–S99), and diseases of the nervous system (G00–G99) were the most common. 

### 2.4. Sociodemographic, Employment, and Social Security Benefits Data

Age at baseline was divided into five-year groups. Information on major geographical region, education (1—tertiary; 2—secondary, and 3—primary), and type of owner (1—private, domestic; 2—private, foreign-owned; 3—public; and 4—unknown) at baseline was obtained from the FLEED. 

Physical heaviness of work was estimated for each occupation using a gender-specific job exposure matrix (JEM) developed earlier in a large population survey [25].

Information on episodes of employment, unemployment, and vocational rehabilitation during the two years preceding baseline was obtained from the FCP registers. Data on sickness absence (SA) spells and days during the two years preceding baseline were obtained from the SII register. Information on partial and temporary work disability retirement during the two years preceding baseline was obtained from the FCP and SII registers. 

We calculated annually the number of days in full work duties, in unemployment, on full SA, with partial work disability (receiving either partial sickness benefit or partial disability pension), participating in vocational rehabilitation, and receiving temporary full disability pension before baseline.

### 2.5. Statistical Analysis

We applied PS matching to control for compositional and attrition bias induced by structural changes in the workforce over time. We obtained matched triplets for observational the years 2007, 2010, and 2013. First, the matching was done for 2007 and 2010. For successfully matched subjects, further matching was done for 2013. To compute PS, we conducted a set of hierarchical logistic regressions with covariates obtained from the registers and assignment to the specific observational year as a dependent variable. The analyses were performed within six subgroups of gender and education. We used the following set of covariates: age, major geographical region (dummy variable), employment sector (dummy variable), number of days in full work duties in year 1 and 2 (ordinal variable), number of days in unemployment in year 1 and 2 (ordinal variable), number of full sickness absence days in year 1 and 2 (ordinal variable); number of partial work disability days in year 1 and 2 (dichotomous variable: none, at least one), number of days on vocational rehabilitation in year 1 and 2 (dichotomous variable: none, at least one), and heavy physical work (dichotomous variable: exposed, non-exposed). The fitness of the model was assessed with the Hosmer and Lemeshow test. PS was used to match individuals on the probability that they would belong to the specific observational year. We applied 1:1 nearest neighbour matching without replacement to minimize conditional bias. We ran several matching models with different caliper values to maximize the number of matched pairs. We set 0.02 as the maximum tolerance for matching.

Using cause-specific sickness absence history during the two years preceding baseline, we also counted the number of different diagnostic groups among an individual as a proxy for comorbidity. There were no differences in either gender in the number of different diagnostic groups between the observational years. We used the comorbidity measure among the potential explanatory variables in the intermediate models for PS calculation, but it was left out from the final model. 

We used generalized estimating equations (GEE) to estimate the age-adjusted one-year cumulative incidence of disability retirement in the observational years 2007, 2010, and 2013. We calculated the changing risk of disability retirement due to all causes, musculoskeletal diseases, and mental disorders for each occupational group separately. For this, we used the Cox regression model to estimate proportional hazards ratios (HR) with 95% confidence intervals (95% CI) for the observational years 2007 and 2013, with 2010 being the reference. The HRs and their 95% CI were adjusted for baseline age and covariates, which were imbalanced after PS matching.

To explore occupational class differences in disability retirement during each observational year, we estimated the risk of disability retirement in each occupational group with upper-level non-manual employees as the reference group. 

All analyses were run separately for men and women. SAS statistical software version 9.4 (SAS Institute Inc., Cary, NC, USA) was used to conduct the analyses.

## 3. Results

### 3.1. Changes in the Composition of the Finnish Workforce over the Period 2007–2013

The employment rate among 30–59-year-old persons on the last day of the year 2006, 2009, and 2012 was 75.3%, 74.6%, and 74.7%, respectively. The rates were higher among women than men. We explored the types of selection out of the workforce among 30–59-year-old men and women, who were employed or self-employed according to their main economic activity by the beginning of the year 2006, 2009, or 2012 (one year preceding the observational year). More men than women left work by the end of each year (Figure 1). Overall, during 2009, the largest decrease in the proportion of working men was seen among skilled manual workers, followed by unskilled manual workers. The proportion of working women generally remained stable in 2006, 2009, and 2012. Nevertheless, similarly to men, about 6% of women in skilled manual occupations had left working life by the end of 2009, with the most common path being transition to unemployment. In both genders, the dropouts due to unemployment increased from 2006 to 2009 and slightly decreased from 2009 to 2012, particularly in manual occupations.

Between 2007 and 2013, the proportion of lower level non-manual employees increased in both genders by 3.1 and 1.3 percentage points, respectively (Table 1). In contrast, the proportion of skilled manual workers among men and unskilled manual workers among women decreased by 3.1 and 1.6 percentage points, respectively. Educational attainment and employment in the public sector increased in both genders. Furthermore, both men and women had the highest work participation (more days in full work duties) during the two years preceding the observational year 2010. Changes in the number of preceding full SA days were negligible over the study period. It is noteworthy that in both genders, the share of people with primary education substantially reduced among skilled and unskilled manual workers (Appendix A). 

### 3.2. Disability Retirement Trends by Occupational Class

Of the 1,114,789 maximum possible triplets, 885,807 (79.5%) remained after PS matching. Matching was more successful in men than in women (82.5% vs. 76.5%). Only minor differences remained between the observational years in the characteristics of the study populations after matching (Appendix A). Especially, occupational class differences in education were reduced (Appendix A). The persons lost during matching were younger, and had primary education and higher work participation (more days in full work duties and less days on SA or in unemployment). 

In the original study population, the age-adjusted, one-year cumulative incidence of disability retirement due to all causes, musculoskeletal diseases, and mental disorders declined between 2007 and 2013 (Figure 2A,C; Figure 3A,C; Figure 4A,C) for all occupational classes, but self-employed. In the matched study population, the cumulative incidence of all-cause and cause-specific disability retirement among skilled and unskilled manual workers in 2007 was lower than in the original sample (Figure 2B,D; Figure 3B,D; Figure 4B,D). In these groups, disability retirement sharply increased in 2010 and then declined in 2013.

Further analyses in the matched population showed that the risk of all-cause and cause-specific disability retirement did not differ between 2007 and 2010 for most of the occupational classes (Appendix A). However, the risk of disability retirement was statistically significantly lower in 2013 than in 2010. A continuous linear decline in all-cause and cause-specific disability retirement during the observational years was observed for lower-level non-manual male employees only. 

### 3.3. Changes over Time in Occupational Class Differences in Disability Retirement

In the matched study population, compared with upper-level non-manual employees, the risk of all-cause and cause-specific disability retirement was higher among all occupational classes, except other self-employed. This was observed for all observational years (Figure 5A–F). The largest occupational class differences were found for disability retirement due to musculoskeletal diseases, particularly in 2007 and 2013 (Figure 5C,D). Among unskilled manual workers, the excess risk of all-cause disability retirement (both genders) and disability retirement due to mental disorders (women) followed an upward linear trend (Figure 5A,B,F). In general, the differences in disability retirement between both skilled and unskilled manual workers and upper-level non-manual employees widened during the period of economic stagnation.

## 4. Discussion

We utilized nationwide register-based datasets and propensity score matching in order to explore occupational class differences in disability retirement trends, while accounting for the potential contribution of changes in the composition of the workforce to these trends. We found that the analyses in the original study population and the matched triplets produced different results regarding the estimate of all-cause and cause-specific disability retirement rates, the shape of the trends, as well as the magnitude of the changes. A downward trend of all-cause and cause-specific disability retirements between 2007 and 2013 observed in the original study population for all occupational classes could not be seen in the PS matched population. We found large occupational class differences in disability retirement, particularly in that due to musculoskeletal diseases. The differences in disability retirement between both skilled and unskilled manual workers and upper-level non-manual employees widened during the period of economic stagnation.

Our results on the differences in the estimates of disability retirement between the initial and matched sample for 2007, 2010, and 2013 suggest that the observed decline between 2007 and 2010 in the unmatched sample, to a large extent, could be explained by contemporaneous structural changes in the workforce followed by the recent economic crisis. However, the decline in disability retirement during the following economic stagnation could only partly be attributed to these factors. Our results on the increasing differences in disability retirement between manual workers and upper-level non-manual employees are in line with previous findings of a growing socioeconomic gap in health during a crisis [9,26]. 

An elevated risk of disability retirement due to all causes and musculoskeletal diseases among manual workers has been reported earlier [13,15,17,19,27] and has been attributed to a range of factors including educational level and exposures to unfavourable working conditions [15,27]. In line with these studies, we observed the highest risk of disability retirement in manual occupational groups that involve heavy physical work. Relatively modest and stable socioeconomic differences over time have been reported in disability retirement due to mental disorders [19]. Accordingly, we found that the occupational class differences were less pronounced in disability retirement due to mental disorders than musculoskeletal diseases. 

The disability retirement trends and occupational class differences may reflect the combined effect of selection on occupation and compositional occupational class differences with respect to age, education, work-related factors, and unemployment. Our results showed that more men than women exited from paid employment by the end of each year. The observed gender differences may be explained by differences in educational attainment and employment patterns between men and women. We observed that women predominantly hold a tertiary education degree, and were employed in lower-level non-manual occupations and in the public sector. Men may have been more affected by the crisis, because they were less educated, and more frequently employed in skilled manual jobs and in the private sector. Overall, the employment rates were higher among women than among men. In Finland, despite strong gender segregation in the labour market, gender differences in any, and full-time employment participation are relatively small compared with many other countries [28]. 

In addition, people may change the workplace or occupation. Those in upper non-manual jobs may be able to stay at work despite their health problems, while deteriorating health and working conditions may impair the possibilities of those in manual occupations to perform their work tasks and thereby increase the probability of occupational mobility [29,30]. Economic crisis can accelerate and make these processes more complex, and may thereby lead to health-based selection out of employment via other pathways than disability retirement [31,32]. 

The disability retirement trend may also have been affected by societal reforms aimed at enhancing work participation via prevention of permanent full disability retirement. Earlier, we reported on the potential of part-time sick leave to substantially reduce the risk of full disability retirement [33]. During the economic recession, the numbers of receivers of the partial sickness benefit and vocational rehabilitation continued to increase [34,35]. 

Because of changes in the composition of the workforce, the risk of disability retirement might be overestimated for some occupational classes and underestimated for others. Furthermore, the magnitude or direction of bias may vary with time. Traditional analytical methods may thus be ineffective in controlling bias. In the present study, we applied PS matching to reduce the impact of bias, induced by changes in the workforce, on occupational differences in disability retirement. PS matching creates populations that are exchangeable (similar on all characteristics). Accordingly, the estimated marginal differences in the outcome will be closer to the true marginal differences than those produced by other traditional confounder-adjusted regression approaches [36]. 

### Strengths and Limitations

The strengths of the current study include the large, nationally representative samples of the Finnish working aged population and register-based data providing rich longitudinal information on employment and sociodemographic factors that are not prone to non-response bias. The very large samples allowed us to evaluate occupational class differences in disability retirement trends for both genders separately. Furthermore, we utilized PS matching to control for potential influence of changes in the composition of the workforce. However, the set of potential explanatory variables was limited to the variables available in the registers and did not include health status and lifestyle factors. An examination of the proxy measure for comorbidity based on cause-specific sickness absence history nevertheless suggested that changes in comorbidity did not influence the findings. Although exposure to heavy physical work, provided by a gender-specific job exposure matrix, was included into calculation of the PS, changes in work exposures and labour market conditions were unavailable. These time-varying confounders, together with changing individuals’ illness behaviour during the economic crisis, can underlie the unexplained occupational differences in the disability retirement trends. 

## 5. Conclusions

In occupational epidemiology, structural changes in the workforce should be taken into account when analysing trends in ill-health. Controlling for these changes revealed widening occupational class differences in disability retirement during the period of economic stagnation. Decision makers should not rely on raw statistics on work disability during fluctuating economic circumstances. 

## Figures and Tables

**Figure 1 ijerph-16-01523-f001:**
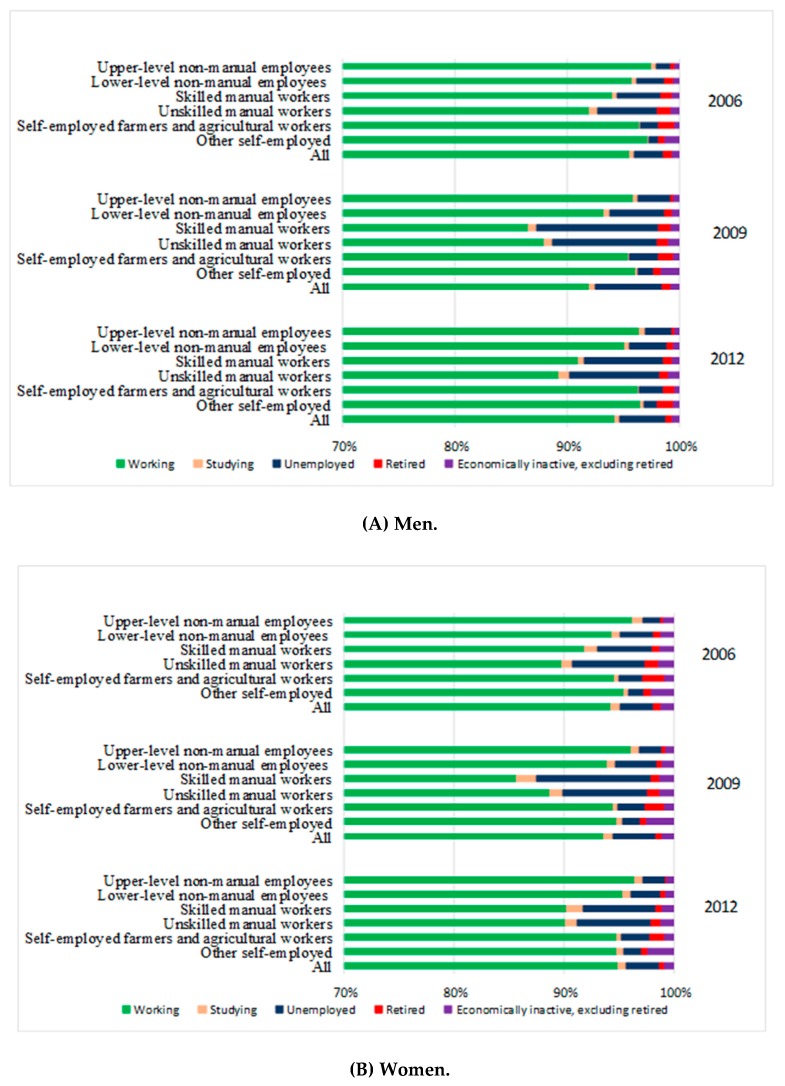
Labor force participation status prior to the baseline by occupational class and observational year among (**A**) men and (**B**) women.

**Figure 2 ijerph-16-01523-f002:**
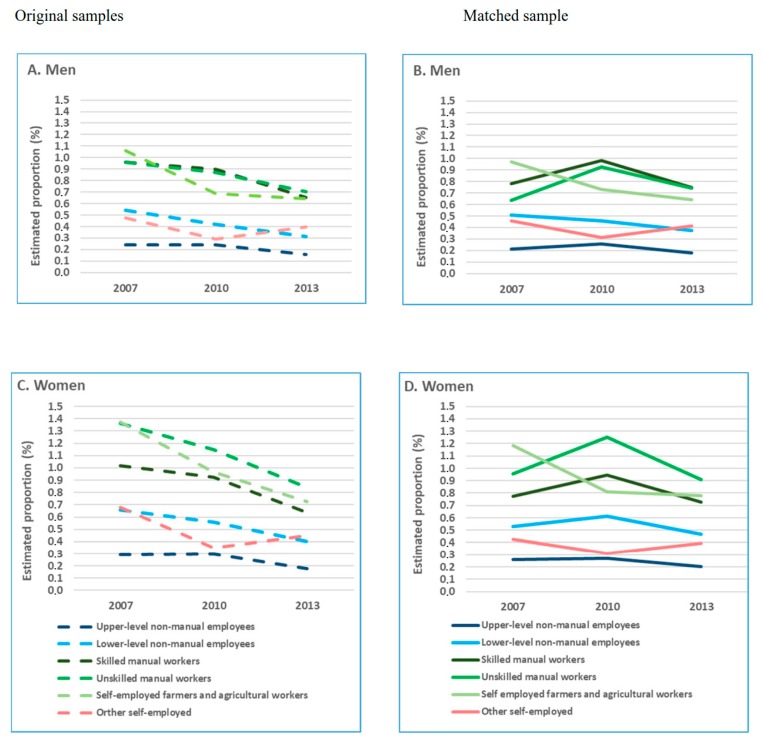
Age-adjusted, one-year cumulative incidence of all-cause disability retirement (%) by occupational class and cause-specific disability retirement (%) before and after propensity score matching among (**A**) men, original samples; (**B**) men, matched samples; (**C**) women, original samples; and (**D**) women, matched samples.

**Figure 3 ijerph-16-01523-f003:**
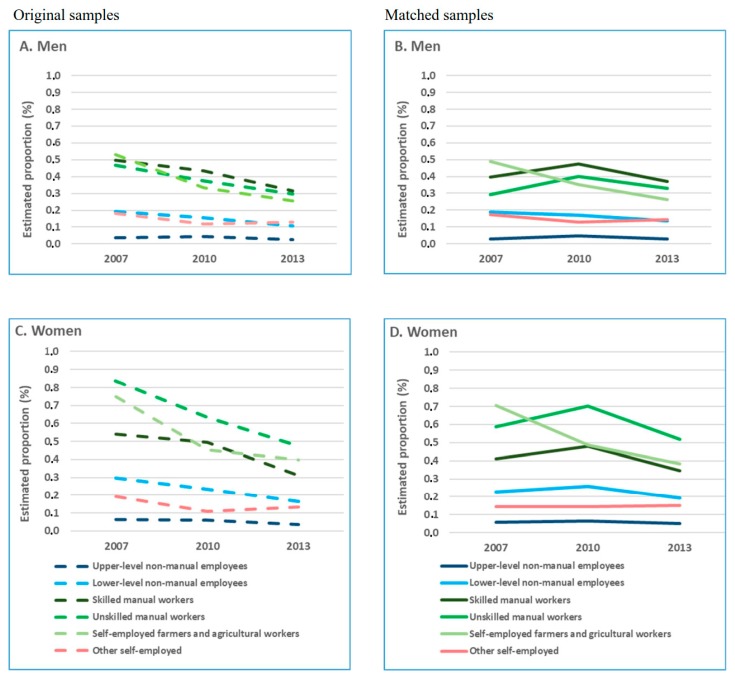
Age-adjusted, one-year cumulative incidence of disability retirement due to musculoskeletal diseases (%) by occupational class and cause-specific disability retirement (%) before and after propensity score matching among (**A**) men, original samples; (**B**) men, matched samples; (**C**) women, original samples; and (**D**) women, matched samples.

**Figure 4 ijerph-16-01523-f004:**
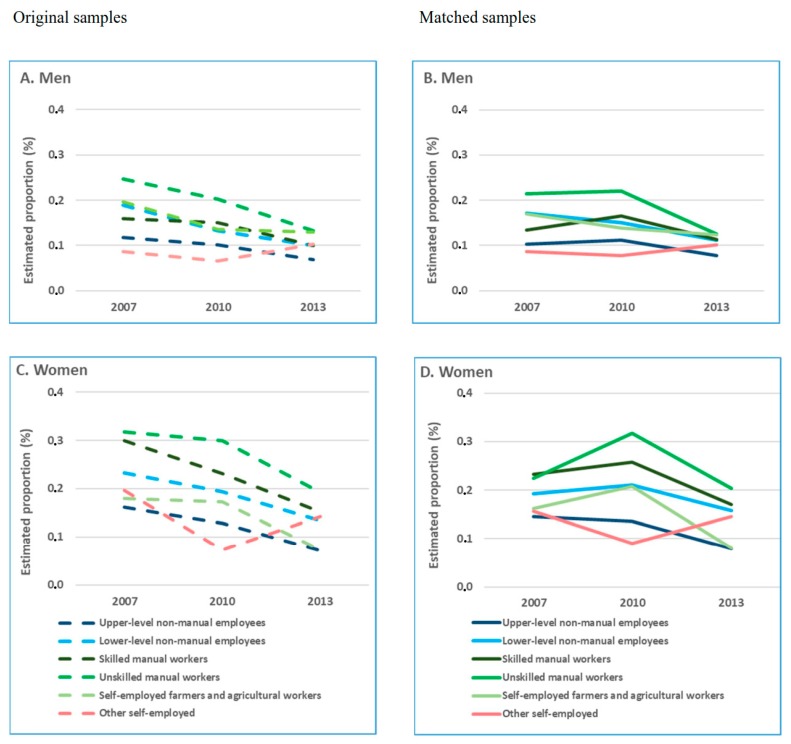
Age-adjusted, one-year cumulative incidence of disability retirement due to mental disorders (%) by occupational class and cause-specific disability retirement (%) before and after propensity score matching among (**A**) men, original samples; (**B**) men, matched samples; (**C**) women, original samples; and (**D**) women, matched samples.

**Figure 5 ijerph-16-01523-f005:**
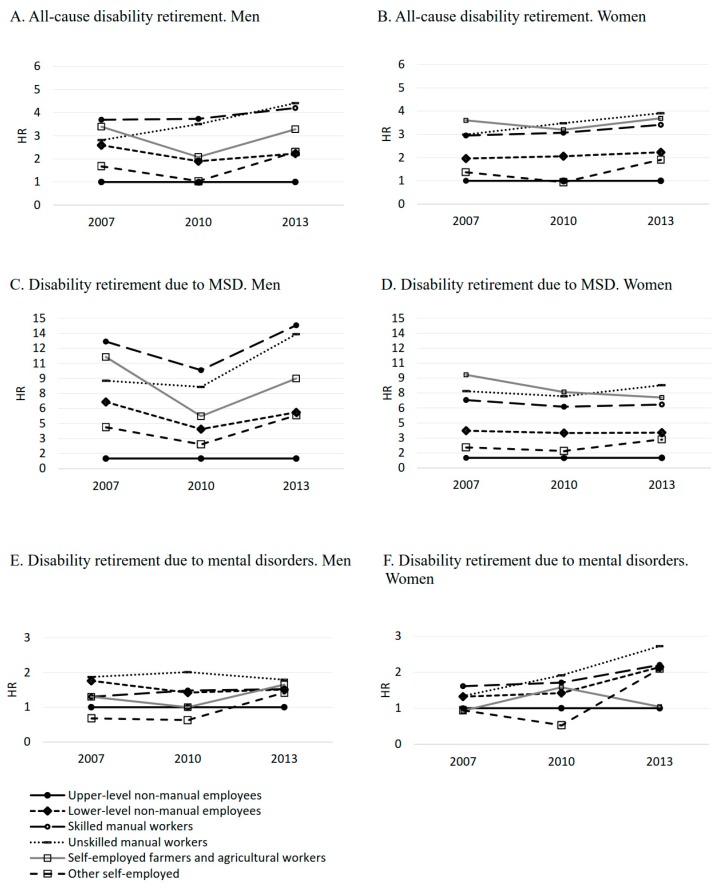
Annual risk (hazards ratios (HRs) adjusted for baseline age) of all-cause and cause-specific disability retirement by occupational class (HR = 1.00 for upper-level non-manual employees) and observational year among 30–59-year-old men and women: (**A**,**B**) all-cause disability retirement, (**C**,**D**) disability retirement due to musculoskeletal diseases (MSD), and (**E**,**F**) disability retirement due to mental disorders.

**Table 1 ijerph-16-01523-t001:** Year-specific distribution (%) of the study population over the study period by selected background characteristics among (A) men and (B) women.

Observational Year	2007	2010	2013	Change (Percentage Points)
*N*	*N*	*N*	2007–2010	2010–2013	2007–2013
(A) Men	585,278	553,162	553,140			
Age						
	30–34	15.5	16.7	16.9	1.2	0.2	1.4
	35–39	16.5	15.5	16.7	−1.0	1.2	0.2
	40–44	18.2	17.6	16.3	−0.6	−1.3	−1.9
	45–49	17.7	18.1	18.0	0.4	−0.1	0.3
	50–54	17.3	17.0	17.1	−0.3	0.1	−0.2
	55–59	14.9	14.9	15.1	0.0	0.2	0.2
Education						
	Tertiary	35.5	37.5	37.9	2.0	0.4	2.4
	Secondary	45.6	46.4	47.4	0.8	1.0	1.8
	Primary	18.9	16.1	14.8	−2.8	−1.3	−4.1
Occupational class						
	Upper-level non-manual employees	23.4	25.1	22.7	1.7	−2.4	−0.7
	Lower-level non-manual employees	23.0	24.2	26.1	1.2	1.9	3.1
	Skilled manual workers	31.6	28.7	28.5	−2.9	−0.2	−3.1
	Unskilled manual workers	6.0	6.1	6.4	0.1	0.3	0.4
	Self-employed farmers and agricultural workers	5.2	4.8	4.3	−0.4	−0.5	−0.9
	Other self-employed	10.9	11.1	12.0	0.2	0.9	1.1
Major region						
	Southern Finland	29.6	31.2	31.6	1.6	0.4	2.0
	Western Finland and Åland	22.2	21.5	21.3	−0.7	−0.2	−0.9
	Eastern Finland	25.3	24.9	25.0	−0.4	0.1	−0.3
	Northern Finland	22.9	22.5	22.2	−0.4	−0.3	−0.7
Employment sector						
	Private, domestic	68.4	67.0	66.9	−1.4	−0.1	−1.5
	Private, foreign	9.7	9.7	9.2	0.0	−0.5	−0.5
	Public	21.9	23.4	23.9	1.5	0.5	2.0
Physically heavy work	32.5	31.0	29.4	−1.5	−1.6	−3.1
Mean days in full work duties						
−2 year	242	296	272			
−1 year	238	299	269			
Mean days on full sickness absence						
−2 year	3.5	3.1	3.1			
−1 year	2.6	2.5	2.6			
Mean days in unemployment						
−2 year	5.6	9.5	6.7			
−1 year	9.6	7.5	11.1			
(B) Women	571,852	566,767	561,649			
Age						
	30–34	13.5	14.9	15.1	1.4	0.2	1.6
	35–39	15.5	14.5	14.7	−1.0	0.2	−0.8
	40–44	18.1	17.4	17.0	−0.7	−0.4	−1.1
	45–49	18.4	18.7	18.6	0.3	−0.1	0.2
	50–54	18.5	18.2	18.1	−0.3	−0.1	−0.4
	55–59	16.0	16.4	16.4	0.4	0.0	0.4
Education						
	Tertiary	47.1	50.6	53.2	3.5	2.6	6.1
	Secondary	39.4	39.1	38.6	−0.3	−0.5	−0.8
	Primary	13.5	10.4	8.2	−3.1	−2.2	−5.3
Occupational class						
	Upper-level non-manual employees	21.5	23.3	23.3	1.8	0.0	1.8
	Lower-level non-manual employees	55.6	55.8	56.9	0.2	1.1	1.3
	Skilled manual workers	5.6	4.6	4.6	1.0	0.0	−1.0
	Unskilled manual workers	8.7	8.1	7.1	−0.6	−1.0	−1.6
	Self-employed farmers and agricultural workers	2.7	2.3	2.2	−0.4	−0.1	−0.5
	Other self-employed	5.9	5.8	5.9	−0.1	0.1	0.0
Major region						
	Southern Finland	31.5	31.9	32.3	0.4	0.4	0.8
	Western Finland and Åland	22.0	21.8	21.6	−0.2	−0.2	−0.4
	Eastern Finland	24.1	24.0	24.1	−0.1	0.1	0.0
	Northern Finland	22.4	22.2	22.0	−0.2	−0.2	−0.4
Employment sector						
	Private, domestic	46.4	45.1	44.5	−1.3	−0.6	−1.9
	Private, foreign	9.3	9.0	8.6	−0.3	−0.4	−0.7
	Public	44.3	45.9	46.9	1.6	1.0	2.6
Physically heavy work	24.4	23.8	22.0	−0.6	−1.8	−2.4
Mean days in full work duties						
−2 year	237	304	280			
−1 year	230	300	274			
Mean days on full sickness absence						
−2 year	4.6	4.1	4.0			
−1 year	3.6	3.6	3.5			
Mean days in unemployment						
−2 year	8.9	9.8	8.5			
−1 year	13.6	12.7	12.8

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
