# Peer review of "Controlling for Structural Changes in the Workforce Influenced Occupational Class Differences in Disability Retirement Trends"

_ijerph, 2019, doi:10.3390/ijerph16091523_

Round 1

Reviewer 1 Report

This epidemiological study is thorough in terms of background literature review, methods data analysis, and discussion. Issues that would improve the overall manuscript could be:

The article becomes very technical throughout with statistical nomenclature and graphs. I think what's important here is a way to make the message clearer: does actually deifferent class lead to the difference in disability retirement? I think the title should reflect this.

The methods are satisfactory.

The graphs of the results could be improved with a slightly enlarged font

Other confounders such as disease (cancer for example or accidents) have not been analyzed. Could this lead to disability?

Can the authors mention why this method was chosen instead of Poisson regression or panel data regression analysis for example?

Discussion: needs more in-depth discussion of other confounders

Author Response

We thank the reviewers for their comments to our submitted manuscript and have revised our paper accordingly. We believe that that the comments helped us to improve the manuscript.

All changes have been marked with track changes in the revised version. Please find below point-by-point responses to the reviewers’ comments.

Reviewer(s)' Comments to Author:

Reviewer: 1

Comments to the Author

This epidemiological study is thorough in terms of background literature review, methods data analysis, and discussion. Issues that would improve the overall manuscript could be:

The article becomes very technical throughout with statistical nomenclature and graphs. I think what's important here is a way to make the message clearer: does actually deifferent class lead to the difference in disability retirement? I think the title should reflect this.

Authors’ response: Thank you for this comment. We revised the title so that it better reflects the message of the paper.

The methods are satisfactory.

The graphs of the results could be improved with a slightly enlarged font

Authors’ response: All figures were revised according to reviewer suggestion. 

Other confounders such as disease (cancer for example or accidents) have not been analyzed. Could this lead to disability?

Authors’ response: Using cause-specific sickness absence history during preceding two years before baseline we counted the number of different diagnostic groups that an individual had. We then used the number of different diagnostic groups as a proxy for comorbidity. There were no differences in either gender in the number of different diagnostic groups between the observational years. Even though the co-morbidity variable was included into one of the intermediate models of PS calculation, it was left out from the final model. We now describe this in the Methods. Cancer and accidents can lead to work disability, and these causes are included in our outcome of all-cause disability retirement. We now list the most common causes of disability retirement in the methods section to clarify the issue (page 7). 

Can the authors mention why this method was chosen instead of Poisson regression or panel data regression analysis for example?

Authors’ response: We assumed that PS matched triplets represent correlated data arising from the similarities (e.g. by education) of individuals within the same occupational class, rather than several observations in the same individual. We used generalized estimating equations to estimate age-adjusted one-year cumulative incidence of disability retirement since it is an increasingly used method to analyze correlated longitudinal data. The panel data regression analysis to examine the changes in disability retirement by occupational class could not be used since our panel data consisted of only 18 observations (6 occupational classes over 3 years).  

Discussion: needs more in-depth discussion of other confounders

Authors’ response: We extended discussion on confounders in the Strengths and limitations part (pages 20-21).

Reviewer 2 Report

I found the article very interesting and providing an important insight to the explanation of disability trends by occupational class in the period of economic crisis and after. The overall quality of the article is high. 

There are some minor issues that I would suggest to change/revise. In the abstract, the first two sentences are somewhat repetitive, it would be good to rephrase. On page 4, line 155, I suggest rephrasing 'gender x education strata' with less jargon explanation. In line 161, I understand that the heavy physical work is also a dummy variable. In line 181, there is an interesting statement that women have higher employment rates compared to men. This is not the case in many of the countries. Maybe it would be good to include the part-time work dummy in the model, as I could guess that women work more part-time and their exposure to work conditions is limited. It would be also good to discuss the observed gender differences in the structure of employment - women work more frequently as lower-level non-manual employees and less as skilled manual workers compared to men. Furthermore, there are differences in the types of sectors that women are employed. It would be interesting to add the sector of employment to the model specification. In Figure 5 (page 11) I could not see the confidence intervals indicated in the figure title. In the conclusions, it would be worth to expand on the policy recommendations for occupational health policy. 

Author Response

We thank the reviewers for their comments to our submitted manuscript and have revised our paper accordingly. We believe that that the comments helped us to improve the manuscript.

All changes have been marked with track changes in the revised version. Please find below point-by-point responses to the reviewers’ comments.

Reviewer(s)' Comments to Author:

Reviewer: 2

Comments to the Author

I found the article very interesting and providing an important insight to the explanation of disability trends by occupational class in the period of economic crisis and after. The overall quality of the article is high.

There are some minor issues that I would suggest to change/revise.

In the abstract, the first two sentences are somewhat repetitive, it would be good to rephrase.

Authors’ response: The second sentence of the abstract has been modified.

On page 4, line 155, I suggest rephrasing 'gender x education strata' with less jargon explanation.

Authors’ response: The text has been revised: “….within six subgroups of gender and education”

In line 161, I understand that the heavy physical work is also a dummy variable.

Authors’ response: Heavy physical work was a dichotomized variable with categories exposed and non-exposed. We have clarified this in the methods.

In line 181, there is an interesting statement that women have higher employment rates compared to men. This is not the case in many of the countries. Maybe it would be good to include the part-time work dummy in the model, as I could guess that women work more part-time and their exposure to work conditions is limited. It would be also good to discuss the observed gender differences in the structure of employment - women work more frequently as lower-level non-manual employees and less as skilled manual workers compared to men. Furthermore, there are differences in the types of sectors that women are employed. It would be interesting to add the sector of employment to the model specification.

Authors’ response: We appreciate this comment since there is large gender segregation in the labor market that further varies between countries. Unfortunately, the registers did not include information on type of employment (part-time or full-time). In Finland, however, full time work is relatively common also among women. Sector of employment does nevertheless vary considerably between the genders, and this was included into the model for calculation of propensity score. We added discussion on gender differences in the structure of employment to the discussion (pages 19-20).

In Figure 5 (page 11) I could not see the confidence intervals indicated in the figure title.

Authors’ response: Confidence intervals were removed from the Figure 5 because they were indistinguishable. The figure title has been revised.

In the conclusions, it would be worth to expand on the policy recommendations for occupational health policy. 

Authors’ response:  We added the following sentence to the conclusion: “Decision makers should not rely on raw statistics on work disability during fluctuating economic circumstances” (page 21).

Reviewer 3 Report

The paper examines how the retirement rate of disability persons in several occupational class is determined at the economic recession years in Finland. The paper shows the detail data related to the unemployment rate, retirement rate and others.

I think that the paper shows the detail data but should be sufficiently examined by empirical analysis. For instance, I will try to set the dependent variable as the retirement of disability persons, the explanatory variables and examine the difference between the retirement rate of not disability persons and disability persons with Difference in Difference analysis. The recession years are considered as the dummy variable. I think that the paper should be considered the abovementioned setting and should be derived the significance with empirical analysis.

In this version of the paper, I can not decide the value of the contribution derived by this paper.

Author Response

We thank the reviewers for their comments to our submitted manuscript and have revised our paper accordingly. We believe that that the comments helped us to improve the manuscript.

All changes have been marked with track changes in the revised version. Please find below point-by-point responses to the reviewers’ comments.

Reviewer(s)' Comments to Author:

Reviewer: 3

Comments to the Author

The paper examines how the retirement rate of disability persons in several occupational class is determined at the economic recession years in Finland. The paper shows the detail data related to the unemployment rate, retirement rate and others.

I think that the paper shows the detail data but should be sufficiently examined by empirical analysis. For instance, I will try to set the dependent variable as the retirement of disability persons, the explanatory variables and examine the difference between the retirement rate of not disability persons and disability persons with Difference in Difference analysis. The recession years are considered as the dummy variable. I think that the paper should be considered the abovementioned setting and should be derived the significance with empirical analysis.

In this version of the paper, I can not decide the value of the contribution derived by this paper.

Authors’ response: We thank the reviewer for the suggestion of analytical strategy. Study population of this nationwide register-based study is composed of three longitudinal cohorts. Each cohort represented the general population and consisted of 30-59 years old persons, who were employed according to their main economic activity at baseline (i.e. 2006, 2008 or 2012). We aimed to estimate the incidence of disability retirement for year 2007, 2010 and 2013 in the general population and explore whether the occupational class differences in disability retirement trend could be explained by structural changes in workforce composition. We therefore do not compare retirement between the disabled and non-disabled. Due to our study design and aim, the suggested analysis cannot be applied. We believe, that analytical methods used in this study are optimal to fulfill the study aims. As suggested by another reviewer, we have revised the title so that it better reflects the content of the paper. We also specify in the Abstract and in the Study population section that general population data are used.

Round 2

Reviewer 1 Report

The author have now addressed my concerns and I am happy with present manuscript 

Author Response

Thank you for your comment.

Reviewer 3 Report

Dear authors

Thank you for the revised paper. 

I can understand the aim of the paper. 

However, I think that it is important to compare retirement between the disabled and non-disabled to make clear the trend of retirement of the disabled. 

I can not decide the quality of this paper without the abovementioned analysis. 

So, I would like to obey the decision of editor of this journal. 

Thank you.

Best regards

Author Response

Authors reply: 

The outcome in our study was disability retirement. In Finland disability pension is granted for Finnish residents with a chronic illness, disability or injury that has been verified by a physician with a medical certificate and evaluated as causing considerable and long-lasting (about 1 year) decreased work ability. Before that, sickness allowance is usually paid for a maximum of 300 working days. Persons who do not fulfill such criteria cannot receive a disability pension. Because the non-disabled would not be eligible for this type of pension, we cannot compare retirement between the disabled and non-disabled. Furthermore, the registers did not include information on health status or grade of disability.